# Altered Protein Kinase A-Dependent Phosphorylation of Cav1.2 in Left Ventricular Myocardium from *Cacna1c* Haploinsufficient Rat Hearts

**DOI:** 10.3390/ijms252413713

**Published:** 2024-12-22

**Authors:** David Königstein, Hauke Fender, Jelena Plačkić, Theresa M. Kisko, Markus Wöhr, Jens Kockskämper

**Affiliations:** 1Institute of Pharmacology and Clinical Pharmacy, Faculty of Pharmacy, Biochemical and Pharmacological Center (BPC) Marburg, University of Marburg, 35032 Marburg, Germany; david.koenigstein@staff.uni-marburg.de (D.K.); haukefender@gmx.de (H.F.); jelenaplackic@yahoo.com (J.P.); 2Center for Mind, Brain and Behavior (CMBB), University of Marburg, 35032 Marburg, Germany; theresa.kisko@kuleuven.be (T.M.K.); woehrm@staff.uni-marburg.de (M.W.); 3Behavioral Neuroscience, Experimental and Biological Psychology, University of Marburg, 35032 Marburg, Germany; 4KU Leuven, Faculty of Psychology and Educational Sciences, Research Unit Brain and Cognition, Laboratory of Biological Psychology, Social and Affective Neuroscience Research Group, B-3000 Leuven, Belgium; 5KU Leuven, Leuven Brain Institute, B-3000 Leuven, Belgium

**Keywords:** *CACNA1C*, L-type calcium channel, Cav1.2, ventricular myocytes, sympathetic stimulation

## Abstract

*CACNA1C* encodes the α1c subunit of the L-type Ca^2+^ channel, Cav1.2. Ventricular myocytes from haploinsufficient *Cacna1c* (*Cacna1c^+/−^*) rats exhibited reduced expression of Cav1.2 but an apparently normal sarcolemmal Ca^2+^ influx with an impaired response to sympathetic stress. We tested the hypothesis that the altered phosphorylation of Cav1.2 might underlie the sarcolemmal Ca^2+^ influx phenotype in *Cacna1c^+/−^* myocytes using immunoblotting of the left ventricular (LV) tissue from *Cacna1c^+/−^* versus wildtype (WT) hearts. Activation of cAMP-dependent protein kinase A (PKA) increases L-type Ca^2+^ current and phosphorylates Cav1.2 at serine-1928. Using an antibody directed against this phosphorylation site, we observed elevated phosphorylation of Cav1.2 at serine-1928 in LV myocardium from *Cacna1c^+/−^* rats under basal conditions (+110% versus WT). Sympathetic stress was simulated by isoprenaline (100 nM) in Langendorff-perfused hearts. Isoprenaline increased the phosphorylation of serine-1928 in *Cacna1c^+/−^* LV myocardium by ≈410%, but the increase was significantly smaller than in WT myocardium (≈650%). In conclusion, our study reveals altered PKA-dependent phosphorylation of Cav1.2 with elevated phosphorylation of serine-1928 under basal conditions and a diminished phosphorylation reserve during β-adrenergic stimulation. These alterations in the phosphorylation of Cav1.2 may explain the apparently normal sarcolemmal Ca^2+^ influx in *Cacna1c^+/−^* myocytes under basal conditions as well as the impaired response to sympathetic stimulation.

## 1. Introduction

L-type calcium (Ca^2+^) channels (LTCCs) play essential roles in neurons and cardiac myocytes, in particular the subtype Cav1.2, which is encoded by the *CACNA1C* gene [1]. Mutations and polymorphisms of *CACNA1C* are implicated in neuropsychiatric and cardiac diseases [1,2,3]. *Cacna1c* haploinsufficient (*Cacna1c^+/−^*) rats serve as a model to study the role of Cav1.2 in the aforementioned diseases. The animals exhibit both a behavioral as well as a cardiac phenotype. The behavioral phenotype includes deficits in pro-social communication and other behavioral alterations (e.g., increased self-grooming behavior) consistent with a putative role of Cav1.2 in several neuropsychiatric disorders [4,5]. The cardiac phenotype was unraveled only recently [6]. Isolated ventricular myocytes from *Cacna1c^+/−^* rats showed unaltered sarcolemmal Ca^2+^ influx, Ca^2+^ transients, and contractions under basal conditions, despite the altered expression or phosphorylation of major Ca^2+^-regulating proteins including Cav1.2, the sarco-/endoplasmic reticulum calcium ATPase (SERCA2a), the sodium/calcium exchanger (NCX), and the ryanodine receptor (RyR2). Following the stimulation of β-adrenergic receptors by isoprenaline, however, myocytes exhibited an attenuated ability to increase sarcolemmal Ca^2+^ influx, Ca^2+^ transients, and contractions. Thus, hearts and myocytes from *Cacna1c^+/−^* rats show an impaired sympathetic stress response. The role of Cav1.2 in this impaired sympathetic stress response in cardiac myocytes from *Cacna1c^+/−^* rats, however, remains elusive.

During sympathetic stimulation of the heart, β-adrenergic receptors become activated by norepinephrine. The resulting signaling cascade leads to an increase in cAMP concentration with subsequent activation of protein kinase A (PKA). PKA activation, in turn, leads to the phosphorylation of Ca^2+^ handling proteins mediating the positive-inotropic effect of sympathetic stimulation in ventricular myocardium [7,8]. A large increase in the L-type Ca^2+^ current (conducted by Cav1.2) is key to this positive-inotropic effect [9,10]. The increase in L-type Ca^2+^ current elicited by β-adrenergic stimulation is PKA-dependent [9,10,11]. PKA-dependent stimulation of the cardiac L-type current occurs mainly through the phosphorylation of Rad, a small GTPase protein associated with the channel complex [7]. PKA, however, also phosphorylates Cav1.2 directly. Several PKA phosphorylation sites have been identified in Cav1.2 [7,11]. The first one identified was serine-1927 (S1927; in mouse and guinea-pig) or serine-1928 (S1928; in rat, rabbit, and human) [11]. S1928 is heavily phosphorylated in rat ventricular myocytes upon β-adrenergic stimulation [12]. Moreover, phosphospecific antibodies targeted against this site are available, enabling direct monitoring of the PKA-dependent phosphorylation of Cav1.2. Here, we used such a phosphospecific antibody against S1928 in order to characterize the PKA-dependent phosphorylation of Cav1.2 and its potential role in the impaired sympathetic stress response of *Cacna1c^+/−^* rats. Our results revealed that ventricular myocardium from *Cacna1c^+/−^* rats (compared to wildtype (WT) rats) exhibited increased phosphorylation of S1928 under basal conditions but an attenuated increase in the phosphorylation of S1928 during β-adrenergic stimulation with isoprenaline. The results suggest that altered PKA-dependent regulation of Cav1.2 contributes to the normal sarcolemmal Ca^2+^ influx under basal conditions as well as to the impaired sympathetic stress response in *Cacna1c^+/−^* rats.

## 2. Results

### 2.1. Expression of Cav1.3 and Cavβ2 in Cacna1c^+/−^ Left Ventricular Myocardium

Despite reduced expression of Cav1.2 in the left ventricular (LV) myocardium of Cacna1c^+/−^ rats, basal L-type Ca^2+^ currents in ventricular myocytes were found to be unchanged [6]. This might be explained by several scenarios. For example, the reduced expression of Cav1.2 might have been compensated for by elevated expression of other voltage-dependent Ca^2+^ channels with similar properties. Alternatively, the regulation of Cav1.2 by phosphorylation could have been changed such that reduced expression of Cav1.2 was compensated for by higher basal phosphorylation levels.

First, we tested whether increased expression of the Cav1.3 channel, also found in the heart [13], might have compensated for the reduced expression of Cav1.2. As shown in Figure 1A,B, expression of Cav1.3 was found in LV myocardium from both WT (+/+) and Cacna1c^+/−^ (+/−) rats. There was, however, no increase in Cav1.3 expression in LV myocardium of Cacna1c^+/−^ rats, but rather an ≈25% decrease compared to WT controls. Moreover, the expression of the regulatory subunit of the Cav1.2 channel, Cavβ2, was unaltered (Figure 1C,D).

### 2.2. Phosphorylation of Cav1.2 at S1928

#### 2.2.1. Detection of Cav1.2 Phosphorylation at S1928

Next, we aimed at evaluating whether the phosphorylation of Cav1.2 in LV myocardium of *Cacna1c^+/−^* rats was altered. Cav1.2 contains several phosphorylation sites. S1928 is a site primarily phosphorylated by PKA upon stimulation of β-adrenergic receptors. Before testing the phosphorylation of this site in *Cacna1c^+/−^* LV myocardium, we first evaluated the ability of the antibody used to reliably detect changes in the PKA-dependent phosphorylation of S1928. To this end, we employed Langendorff-perfused hearts from Sprague-Dawley rats treated with two different solutions. One solution was designed to induce high levels of PKA-dependent phosphorylation. It contained (1) isoprenaline, a β-adrenergic agonist, stimulating increases in cAMP concentration and PKA activity; (2) IBMX, an inhibitor of phosphodiesterases; as well as the phosphatase inhibitors (3) cantharidin and (4) cyclosporin A (phosphorylation or P solution). The other solution was designed to induce dephosphorylation. It contained (1) BDM and the protein kinase inhibitors (2) H-89 and (3) KN-62 (dephosphorylation or De solution). Four hearts each were treated with either a phosphorylation or dephosphorylation solution. LV homogenates of these hearts were subjected to standard Western blotting protocols using a rabbit polyclonal antibody directed against phosphorylated S1928 as previously characterized and validated [14].

As shown in Figure 2, LV homogenates from all four hearts treated with a phosphorylation solution (P) showed a clear strong band at a molecular weight of somewhat below 250 kDa. Conversely, there was an absence of any discernible signal for LV homogenates from all four hearts treated with the dephosphorylation solution (De). Thus, this experiment confirms the reliability of the antibody used to detect changes in the PKA-dependent phosphorylation of Cav1.2 at S1928.

#### 2.2.2. Increased Basal Phosphorylation of Cav1.2 at S1928 in Cacna1c^+/−^ LV Myocardium

Having validated the antibody used, we next sought to examine the baseline phosphorylation level of S1928 on the Cav1.2 channel in LV myocardium from *Cacna1c^+/−^* rats. To accomplish this, we initially assessed the total expression of the Cav1.2 channel in *Cacna1c^+/−^* LV myocardium in comparison to WT (+/+) control. As shown in Figure 3A,B, there was a reduction by ≈30% in the total protein expression of Cav1.2 in *Cacna1c^+/−^* LV tissue when compared to the WT control group. This value is similar to the one reported previously [6]. We then determined the phosphorylation of Cav1.2 at S1928 using the same two sets of samples. As illustrated in Figure 3C, the phosphorylation of Cav1.2 at S1928 was increased in *Cacna1c^+/−^* LV tissue. When normalized to GAPDH expression, the phosphorylation levels of S1928 were elevated by ≈50% in the *Cacna1c^+/−^* group (Figure 3D). When normalizing to Cav1.2 expression (which is reduced in *Cacna1c^+/−^*), the phosphorylation of S1928 in LV myocardium from *Cacna1c^+/−^* was roughly doubled (+110%) compared to the WT controls (Figure 3E). These results reveal a greatly increased baseline phosphorylation of Cav1.2 at S1928 in LV myocardium from *Cacna1c^+/−^* rats.

#### 2.2.3. Attenuated Increase in Phosphorylation of Cav1.2 at S1928 in Cacna1c^+/−^ LV Myocardium During Sympathetic Stress

To investigate the impact of sympathetic stress on the phosphorylation of Cav1.2 at S1928, we subjected Langendorff-perfused hearts to treatment with isoprenaline at a concentration of 100 nM, as described previously [6]. Subsequently, LV tissue homogenates from these hearts were tested for expression of Cav1.2 and phosphorylation at S1928 using Western blot analysis. The findings are presented in Figure 4.

A total of 32 hearts were used for this series: 16 WT (+/+) and 16 *Cacna1c^+/−^* (+/−) hearts. Each membrane contained eight samples: four WT (+/+) and four *Cacna1c^+/−^* (+/−) samples, which were applied in pairs of one untreated control (Ctrl) and one isoprenaline-treated (Iso) sample. This enabled direct comparison of the expression of Cav1.2 and the isoprenaline-induced increase in S1928 phosphorylation between WT and *Cacna1c^+/−^* on each membrane. Data were normalized to WT controls. Our initial step involved the assessment of total Cav1.2 expression levels (Figure 4A). Similar to our observations under basal conditions (Figure 3A,B), we noted a reduction in the expression of the Cav1.2 channel in *Cacna1c^+/−^* LV myocardium by ≈30% (Figure 4A,B).

Using the same set of samples and their respective positions on the membrane, we investigated the phosphorylation of S1928 on the Cav1.2 channel (Figure 4C). While no significant differences in phosphorylation were observed in the untreated control groups of both genotypes, there was a substantial disparity in the phosphorylation status of Serine-1928 in the isoprenaline-treated groups. When normalized to GAPDH expression, the isoprenaline-induced increase in S1928 phosphorylation amounted to ≈590% in the WT group, whereas in the *Cacna1c^+/−^* group, the increase was only ≈280% (Figure 4D). To account for the reduced expression of the Cav1.2 channel in the *Cacna1c^+/−^* rats, the phosphorylation status of S1928 was also normalized to Cav1.2 expression (Figure 4E). Again, there were no significant differences between the untreated control groups of both genotypes. Using this kind of analysis, isoprenaline increased S1928 phosphorylation by ≈650% in LV myocardium from WT rats. In contrast, the isoprenaline-induced effect was significantly attenuated in LV myocardium from *Cacna1c^+/−^* rats, where the phosphorylation at S1928 increased by only ≈410%.

## 3. Discussion

Haploinsufficient *Cacna1c^+/−^* rats have proven to be a unique animal model exhibiting both a behavioral and a cardiac phenotype, thus enabling studies of the potential role of *CACNA1C*/Cav1.2 in neuropsychiatric and cardiac diseases [4,5,6]. We have previously characterized the cardiac phenotype of *Cacna1c^+/−^* rats with respect to cellular Ca^2+^ handling and contraction [6]. We found substantial remodeling of the expression and phosphorylation of major Ca^2+^ handling proteins and an impaired response to sympathetic stress. The very role of Cav1.2 for this altered Ca^2+^ handling, however, remained enigmatic. Here, we show that the altered PKA-dependent phosphorylation of Cav1.2 is involved in this altered Ca^2+^ handling both under basal conditions and following sympathetic stress.

### 3.1. Potential Role of Altered Cav1.2 Phosphorylation for Basal Ca^2+^ Handling in Ventricular Myocardium from Cacna1c^+/−^ Rats

Previous studies in mouse models with heterozygous knockout of *Cacna1c* in the heart have shown that, despite reduced expression of Cav1.2, the L-type Ca^2+^ current in the cardiac myocytes remains rather unaltered or is reduced much less than would be expected from the degree of downregulation of the mRNA or Cav1.2 protein [15,16]. This implies that cardiac myocytes are able to find ways to maintain a certain amount of functional Cav1.2 channels in the sarcolemma, which is required for proper function, i.e., Ca^2+^ transients and contractile activity. It is not always clear, though, how this is achieved.

In our initial cardiac characterization of the *Cacna1c^+/−^* rat model [6], we observed an ≈30% reduction in the expression of Cav1.2 in LV myocardium compared to WT littermates, and this is confirmed here (Figure 3 and Figure 4). Despite this substantial reduction in the protein amount of Cav1.2, however, the L-type Ca^2+^ current and the sarcolemmal Ca^2+^ influx (mainly carried by L-type Ca^2+^ channels) were essentially unchanged [6]. In order to unravel this conundrum, we considered two different scenarios here. First, we tested for a potential upregulation of auxiliary subunits or other Cav proteins, which might have compensated for the reduced expression of Cav1.2. However, neither the β2 subunit, Cavβ2, nor Cav1.3 was found to be upregulated in LV myocardium from *Cacna1c^+/−^* rats (Figure 1), making the first scenario rather unlikely. Second, we tested for altered PKA-dependent phosphorylation of Cav1.2. Using a well-characterized and validated antibody [14], which reliably and sensitively detects the phosphorylation of Cav1.2 at S1928 (Figure 2), a PKA-dependent phosphorylation site, we found that Cav1.2 exhibits substantially increased (roughly doubled) phosphorylation under baseline conditions in LV myocardium from *Cacna1c^+/−^* rats (Figure 3). Activation of the cAMP-PKA pathway causes a large increase in L-type Ca^2+^ current in ventricular myocytes, mainly by means of an increase in the open probability of individual L-type channels [7]. Thus, the elevated PKA-dependent phosphorylation of Cav1.2 observed in this study in LV myocardium from *Cacna1c^+/−^* rats could explain the unaltered L-type Ca^2+^ current and sarcolemmal Ca^2+^ influx in ventricular myocytes from *Cacna1c^+/−^* rats observed previously in the presence of a reduced expression of Cav1.2: In *Cacna1c^+/−^* ventricular myocytes, there are less Cav1.2 channels, but these are PKA “hyperphosphorylated” under baseline conditions, leading to a higher open probability of the individual channels and, thus, normalizing the global Cav1.2/L-type Ca^2+^ channel-mediated sarcolemmal Ca^2+^ influx. With normal sarcolemmal Ca^2+^ influx, there will be normal Ca^2+^ transients and contractions, as observed [6].

### 3.2. Potential Role of Altered Cav1.2 Phosphorylation for the Impaired Sympathetic Stress Response in Ventricular Myocardium from Cacna1c^+/−^ Rats

Ventricular myocytes from *Cacna1c^+/−^* rats exhibit an impaired response to isoprenaline, i.e., sympathetic stimulation, such that the increases in sarcolemmal Ca^2+^ influx, the Ca^2+^ transient, and sarcomere shortenings upon isoprenaline stimulation are less pronounced than in WT myocytes [6]. The role of altered phosphorylation of Cav1.2 in this impaired response, however, remained elusive. The results of this study convincingly show that isoprenaline-induced, PKA-dependent phosphorylation of Cav1.2 in LV myocardium from *Cacna1c^+/−^* rats is attenuated compared with WT, thus explaining the attenuated increases in sarcolemmal Ca^2+^ influx. With less trigger Ca^2+^, there will be less RyR2-mediated Ca^2+^ release from the SR and, hence, a reduced Ca^2+^ transient and reduced sarcomere shortenings in *Cacna1c^+/−^* ventricular myocytes in the presence of sympathetic stimulation compared to WT myocytes. The impairments in SR Ca^2+^ release (caused by reduced trigger Ca^2+^) will be exacerbated by the impaired phosphorylation of RyR2 observed previously [6].

### 3.3. Altered Phosphorylation of Cav1.2 in Human Cardiac Disease

In ventricular myocytes isolated from human failing hearts (heart failure with reduced ejection fraction, HFrEF), most earlier studies reported an unchanged density of the L-type Ca^2+^ current [17,18,19,20,21], despite some evidence for robust reductions of channel mRNA and dihydropyridine binding sites [22]. Moreover, β-adrenergic stimulation of L-type Ca^2+^ current was blunted in both atrial and ventricular myocytes isolated from human failing hearts [23]. Multiple mechanisms may contribute to the impaired β-adrenergic signaling in human HFrEF including altered PKA-dependent phosphorylation of Cav1.2 (and other Ca^2+^-regulating proteins). Direct evidence from single-channel recordings suggests that under basal conditions, single L-type Ca^2+^ channels exhibited increased availability and open probability and kinetic properties similar to PKA-modified channels [24]. The channel properties resembled those caused by stimulation with a cAMP analogue and could not be further altered by inhibition of protein phosphatases [24]. Whole-cell recordings of L-type Ca^2+^ current revealed no differences in current density between failing and non-failing myocytes but a greatly attenuated stimulation by a cAMP analogue in failing myocytes [25]. Moreover, in failing myocytes, protein phosphatase 2A reduced L-type Ca^2+^ current, whereas the inhibition of protein phosphatases did not affect the current [25]. In addition, the L-type Ca^2+^ current from failing myocytes exhibited a blunted response to stimulation with Bay K8644, a dihydropyridine agonist, which could be normalized by pretreatment with acetylcholine [26]. Collectively, these studies provide strong evidence for the notion that, in ventricular myocytes from human end-stage failing hearts, there is a reduction in the number of L-type Ca^2+^ channels, which exhibit elevated phosphorylation by PKA to increase the open probability of individual channels, thereby normalizing the whole-cell current. Because of this PKA-dependent “hyperphosphorylation” under basal conditions, there is a diminished phosphorylation reserve of the channels during β-adrenergic stimulation, resulting in an attenuated response of the whole-cell L-type Ca^2+^ current. This situation in human HFrEF closely resembles the altered PKA-dependent regulation of Cav1.2 in ventricular myocytes from *Cacna1c^+/−^* rats reported here and in our previous study [6]: *Cacna1c^+/−^* myocytes exhibit a reduced expression of Cav1.2, which is “hyperphosphorylated” by PKA at S1928, presumably to increase the open probability and normalize the whole-cell current under basal conditions. Because of this “hyperphosphorylation” under basal conditions, there is a diminished phosphorylation reserve causing an attenuated increase in whole-cell current during β-adrenergic stimulation. Remarkably, many similarities also exist with regard to the PKA-dependent phosphorylation and regulation of RyR2 between human HFrEF and the *Cacna1c^+/−^* rats: in both cases, there is PKA-dependent “hyperphosphorylation” of RyR2 at S2808/S2809 under basal conditions, and this hyperphosphorylation increases the open probability of the channel [27].

Most recently, the reduced phosphorylation of Cav1.2 at S1928 has been shown to contribute to the reduction of L-type Ca^2+^ current in human atrial fibrillation [28]. The reduced phosphorylation of Cav1.2 in atrial fibrillation was shown to result from increased association of Cav1.2 with PDE8B2 and reduced local cAMP levels in the vicinity of the channel [28]. These findings are noteworthy in several respects: (1) they underscore that the regulation (of the phosphorylation) of Cav1.2 occurs in its immediate vicinity and that Cav1.2 is part of a macromolecular multi-protein complex, as also suggested for the channel in ventricular myocytes. (2) They show that local cAMP concentration and PKA activity in the microenvironment of Cav1.2 is an important regulator of L-type Ca^2+^ channel activity. (3) They demonstrate that altered PKA-dependent phosphorylation of Cav1.2 is involved not only in ventricular but also in atrial pathologies.

### 3.4. An Overview on Altered Cellular Ca^2+^ Handling in Ventricular Myocardium from Cacna1c^+/−^ Rats Under Basal Conditions and During Sympathetic Stress

The results of this as well as our previous study [6] reveal altered cellular Ca^2+^ handling in ventricular myocytes from *Cacna1c^+/−^* rats both under basal conditions as well as during sympathetic stimulation (mimicked by the β-adrenergic agonist isoprenaline). These alterations are summarized and illustrated in Figure 5 for basal conditions and in Figure 6 for β-adrenergic stimulation.

Figure 5 illustrates the situation in a WT myocyte (top) and in a *Cacna1c^+/−^* myocyte (bottom). Ventricular myocytes from *Cacna1c^+/−^* rats exhibit a decreased expression of Cav1.2 (−30%) but increased expression of NCX (+20%) and SERCA2a (+50%). The latter two proteins are responsible mainly for Ca^2+^ extrusion from the cytosol following the systolic Ca^2+^ increase. NCX is distributed in the T-tubules and along the surface sarcolemma [29]. SERCA2a, in turn, localizes to the longitudinal SR [30]. The expression of other major Ca^2+^-regulating proteins is unaltered, including RyR2 and PLB. Major alterations with regard to phosphorylation occur in the dyadic cleft, where clusters of Cav1.2 in the T-tubular membrane and clusters of RyR2 in the junctional SR are localized vis-à-vis [31,32]. Both Cav1.2 and RyR2 are part of macromolecular complexes, which may include protein kinases (e.g., PKA), protein phosphatases (e.g., PP2A), phosphodiesterases (e.g., PDE4, PDE8), and scaffolding or adapter proteins (e.g., A-kinase anchoring proteins) [28,33,34,35]. This organization allows for a very local control of cAMP signaling in the immediate vicinity of the channels and, hence, local control of channel phosphorylation by PKA. In WT myocytes, the phosphorylation of Cav1.2 and RyR2 (as well as PLB) is very low, as depicted by the “empty”, i.e., non-phosphorylated serine (S) residues (white circles). By contrast, in *Cacna1c^+/−^* myocytes, in the absence of β-adrenergic stimulation, there is greatly increased phosphorylation of both Cav1.2 and RyR2 at PKA sites S1928 (+110%) and S2808 (+120%), respectively, as depicted by some phosphorylated (P) residues (yellow circles). The phosphorylation of PLB, which interacts with SERCA in the longitudinal SR [30], is equally low in WT and *Cacna1c^+/−^* myocytes. These findings imply that under basal conditions, local cAMP concentration and PKA activity in the dyadic cleft near Cav1.2 and RyR2 are elevated in *Cacna1c^+/−^* myocytes (note blue cAMP/PKA symbols here), whereas cAMP levels and PKA activity in the bulk cytosol are very low and do not differ between *Cacna1c^+/−^* and WT myocytes. Elevated PKA-dependent phosphorylation increases the open probability of both Cav1.2 and RyR2, thus augmenting Ca^2+^-induced Ca^2+^ release in *Cacna1c^+/−^* myocytes, and this mechanism is proposed to underlie the apparently normal sarcolemmal Ca^2+^ influx and Ca^2+^ transients in *Cacna1c^+/−^* myocytes despite the reduced expression of Cav1.2.

The situation is different, however, during sympathetic stress, as depicted in Figure 6. When β-adrenergic receptors (β-AR) are activated by isoprenaline (ISO), this results in the stimulation of adenylate cyclases (AC), a global increase in cAMP levels and activation of PKA throughout the cytosol. Thus, PKA phosphorylates PLB in the longitudinal SR (at S16) but also Cav1.2 (at S1928) and RyR2 (at S2808) in the dyadic cleft. The PKA phosphorylation of PLB at S16 during isoprenaline stimulation does not differ between *Cacna1c^+/−^* and WT myocytes, suggesting similar cAMP increases in the bulk cytosol in *Cacna1c^+/−^* and WT myocytes. In and near the dyadic cleft, however, isoprenaline-induced PKA-mediated phosphorylation of Cav1.2 and RyR2 differs between WT and *Cacna1c^+/−^* myocytes. In WT myocytes, isoprenaline elicits a full-blown response causing PKA-mediated phosphorylation (yellow circles) of (almost) all available S1928 and S2808 residues in Cav1.2 and RyR2. In *Cacna1c^+/−^* myocytes, on the other hand, isoprenaline also increases the phosphorylation of Cav1.2 and RyR2, i.e., on top of the already elevated PKA phosphorylation under basal conditions, but—importantly—this increase is *less* pronounced than in WT myocytes. Maximal PKA phosphorylation of Cav1.2 and RyR2 is reduced in *Cacna1c^+/−^* compared to WT myocytes, leaving “empty”, i.e., non-phosphorylated serine residues (S1928 and S2808) in Cav1.2 and RyR2 (see white circles) and suggesting—once again—altered local regulation of cAMP and PKA-dependent phosphorylation in the dyadic cleft of *Cacna1c^+/−^* myocytes. Thus, the increased phosphorylation of Cav1.2 and RyR2 in *Cacna1c^+/−^* myocytes under basal conditions comes at the expense of an attenuated response to β-adrenergic stimulation.

### 3.5. Conclusions and Perspectives

In summary, there is increased basal PKA-dependent phosphorylation and a reduced phosphorylation reserve during the sympathetic stress of Cav1.2 at S1928 in ventricular myocytes from *Cacna1c^+/−^* rats. In conjunction with the similarly altered phosphorylation of RyR2 at S2808 and the augmented expression of NCX and SERCA [6], these alterations in cellular Ca^2+^ handling may explain (1) the apparently normal sarcolemmal Ca^2+^ influx and excitation–contraction coupling in the face of reduced expression of Cav1.2 and (2) the impaired response to sympathetic stress in *Cacna1c^+/−^* ventricular myocytes. The PKA-dependent “hyperphosphorylation” of Cav1.2 and RyR2 observed under basal conditions in ventricular myocardium from apparently healthy *Cacna1c^+/−^* rats resembles the altered regulation of Cav1.2 and RyR2 found in human failing hearts. Therefore, we speculate that this kind of remodeling of cellular Ca^2+^ handling may increase the susceptibility of *Cacna1c^+/−^* rats to the development of cardiac disease during periods of physical or psychological stress.

## 4. Materials and Methods

### 4.1. Animals

This study was approved by local animal welfare authorities and was performed in accordance with the European Union Council Directive 2010/63/EU and the German Animal Welfare Act.

The generation of *Cacna1c^+/−^* (+/−) rats and wildtype (WT, +/+) littermates was performed as previously described [6]. Female rats at the age of 9–15 months were used in this study. For some experiments (validation of the anti-phospho-serine-1928 antibody, Figure 2), female Sprague-Dawley rats at the age of 6–8 months were used. These rats were obtained from Charles River (Cologne, Germany). All rats had free access to standard chow and water. For isolation of the hearts, the rats were anesthetized through exposure to isoflurane, and deep anesthesia was verified by the absence of pain reflexes. Subsequently, the rats were euthanized via decapitation using a guillotine. After the animals were sacrificed, their hearts were promptly removed and employed for either tissue processing or attachment to the Langendorff perfusion apparatus.

### 4.2. Left Ventricular (LV) Tissue Isolation

After excision of hearts or use of hearts on the Langendorff apparatus, the hearts were placed in an ice-cold solution. Left ventricles were separated, frozen in liquid nitrogen, and stored at −80 °C. LV tissue lysates were used for investigating protein expression or phosphorylation using immunoblotting (as described in Section 4.4).

### 4.3. Langendorff Perfusion and Treatment of Whole Hearts

To assess protein phosphorylation levels in *Cacna1c^+/−^* and WT samples, whole hearts were perfused on a Langendorff apparatus for 5 min with a 2,3 butanedione monoxime (BDM)-containing Tyrode’s solution composed of (mM) 130 NaCl, 5.4 KCl, 0.5 MgCl_2_, 0.15 CaCl_2_, 0.33 NaH_2_PO_4_, 25 HEPES, 22 glucose, pH 7.4 (with NaOH), and 1 mg/mL (≈9.9 mM) BDM. This initial perfusion with a BDM-containing solution served to depress contractile activity of the heart, thereby sparing cellular ATP, as BDM acts as an inhibitor of the actomyosin ATPase [36]. In addition, it served to induce a standard low level of phosphorylation of cardiac myocyte proteins, as BDM is a chemical phosphatase. Afterwards, hearts were perfused for another 5 min with either control solution (Tyrode’s solution without BDM) or with isoprenaline-containing Tyrode’s solution, from which BDM had been omitted and to which 100 nM isoprenaline had been added. Subsequently, the hearts were sectioned and frozen as described above.

To validate the anti-phospho-S1928 Cav1.2 antibody (phospho-Cav1.2 antibody), the hearts were perfused initially with a BDM-containing Tyrode’s solution as described above. Afterwards, the hearts were perfused either with a solution designed to induce maximal PKA-dependent phosphorylation levels (phosphorylation solution) or with a solution designed to induce maximal dephosphorylation (dephosphorylation solution). The phosphorylation solution was a Tyrode’s solution (without BDM) that contained, additionally, 100 nM isoprenaline, 100 µM 3-isobutyl-1-methylxanthine (IBMX), 30 µM cantharidin, and 1 µM cyclosporin A. The dephosphorylation solution, in turn, was a BDM-containing Tyrode’s solution additionally containing the protein kinase inhibitors H-89 (1 µM) and KN-62 (1 µM). Following perfusion with a phosphorylation (5 min) or dephosphorylation (10 min) solution, the hearts were sectioned and frozen as above.

### 4.4. Immunoblotting (Western Blotting)

Western blotting for the results shown in Figure 1 (expression of Cav1.3 and Cavβ2) was performed essentially as described previously [6]. For the remainder, i.e., the results shown in Figure 2, Figure 3 and Figure 4, we used slight modifications as described in detail in the following text: the left ventricular (LV) tissue was homogenized using micro tissue grinders (Wheaton, UK) with a homogenization buffer comprising a mixture of protease and phosphatase inhibitors (PhosStop™ and cOmplete™ Protease Inhibitor Cocktail, Merck, Darmstadt, Germany). Subsequently, protein concentrations were quantified through a BCA assay (Thermo-Fisher Scientific, Waltham, MA, USA) with a BSA standard curve. Protein expression and phosphorylation in homogenates were assessed using standard immunoblotting (Western blotting). Proteins were separated using SDS-PAGE with precast gradient gels (4–20% Mini-PROTEAN TGX, Bio-Rad, Neuried, Germany). The samples, each containing 30 μg of total protein, were prepared with Laemmli buffer containing 5% β-mercaptoethanol. In order to estimate the molecular weight of the analyzed proteins, 5 μL of PageRuler™ Plus Prestained Protein Ladder (Thermo-Fisher Scientific, Waltham, MA, USA) was loaded alongside the samples. The voltage was initially set at 90 V for 1 h and subsequently increased to 120 V until optimal separation was obtained. Separated proteins were transferred from the gel to a nitrocellulose membrane (0.45 μm, BioRad, Neuried, Germany) through the wet blotting procedure. Electrical current was set to 360 mA per gel for 2 h, and transfer took place in a cooling chamber at 4 °C. After the transfer, membranes were cut between the protein bands of interest, allowing for each segment of the membrane to be individually incubated with the respective antibody of interest. The membranes were subsequently washed with TBST buffer (3 times for 10 min each) on a rocking platform. Blocking was performed for 1 h at room temperature with 5% milk (i.e., skimmed milk powder, Sigma-Aldrich, Taufkirchen, Germany) in TBST (or 5% BSA in TBST, for phospho-Cav1.2). After that, the membranes were incubated with the primary antibody in 5% milk (or 5% BSA, for phospho-Cav1.2) in TBST overnight. The following primary antibodies were used (host; dilution; company, catalogue number): anti-Cav1.2 (CACNA1C) antibody (rabbit, 1:1,000; Alomone, Jerusalem, Israel, #ACC-003), phospho-Cav1.2 (Ser1927) polyclonal antibody (rabbit; 1:1,000; Invitrogen/Thermo-Fisher Scientific, Waltham, MA, USA, #PA5-64748), anti-Cav1.3 (CACNA1D) antibody (rabbit; 1:200; Alomone, #ACC-005), anti-CACNB2 antibody (rabbit; 1:800; Alomone, Jerusalem, Israel, #ACC-105), and anti-GAPDH mouse mAB (6c5) antibody (mouse; 1:50,000; Calbiochem, San Diego, CA, USA, #CB1001). On the following day, the membranes were washed (with TBST, 3 times for 10 min each), followed by incubation with the secondary antibody (in 5% milk (or 5% BSA, for phospho-Cav1.2) in TBST, 1 h, at room temperature) and another round of washing (with TBST, 3 times for 10 min each). The secondary antibodies used were either immunopure goat anti-mouse IgG, peroxidase conjugated (Thermo-Fisher Scientific, Waltham, MA, USA, #31430), or immunopure goat anti-rabbit IgG, peroxidase conjugated (Thermo-Fisher Scientific, Waltham, MA, USA, #31460), with a dilution of 1:5000 in each case. The chemiluminescence reaction was detected using a Chemidoc-XRS system (Bio-Rad, Neuried, Germany) with Quantity One Software (Version 4.6.5) after incubation of the membranes for 1 min with the reagent SuperSignalTM West Pico PLUS Chemiluminescent Substrate (Thermo-Fisher Scientific, Waltham, MA, USA) or the more sensitive SuperSignalTM West Femto Maximum Sensitivity Substrate (Thermo-Fisher Scientific, Waltham, MA, USA). Blots were analyzed using FIJI-2 (NIH, USA, Version 2.14.0/1.54f). GAPDH was used as a loading control. To compare the expression and phosphorylation of Ca^2+^ handling proteins between genotypes, the results were normalized to the averaged WT signal (in the absence of isoprenaline) on a given membrane (=100%).

All original Western blot images are shown in Appendix A.

### 4.5. Statistics

Statistical analysis was performed with GraphPad Prism 9 (GraphPad Software, San Diego, CA, USA). Data are presented as scatter plots with bar graphs indicating mean ± SEM. The number of animals is provided as “N”. Data sets were compared by means of an unpaired, two-tailed Student’s *t* test and considered significant when *p* < 0.05. For multiple group comparisons, ANOVA was applied. Asterisks indicate the following levels of significance: * *p* < 0.05, ** *p* < 0.01, *** *p* < 0.001, **** *p* < 0.0001.

## Figures and Tables

**Figure 1 ijms-25-13713-f001:**
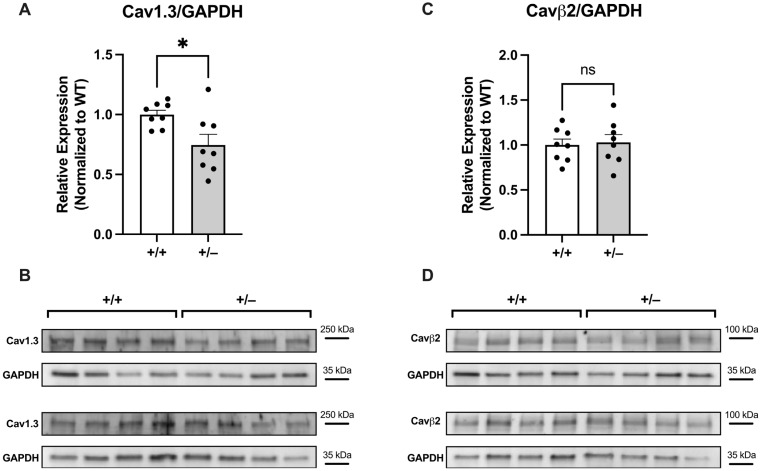
Expression of Cav1.3 and Cavβ2 in LV myocardium from WT and *Cacna1c^+/−^* rats. (**A**) Expression of Cav1.3 in LV myocardium from WT and *Cacna1c^+/−^* rats. Values were normalized to the mean from WT. (**B**) Original Western blot images of 8 WT (+/+) and 8 *Cacna1c^+/−^* (+/−) samples derived from two membranes. The protein used for normalization (GAPDH) derived from the same membrane is shown below the protein of interest. (**C**) Expression of Cavβ2 in LV myocardium from WT and *Cacna1c^+/−^* rats. Values were normalized to the mean from WT. (**D**) Original Western blot images of 8 WT (+/+) and 8 *Cacna1c^+/−^* (+/−) samples derived from two membranes. The protein used for normalization (GAPDH) derived from the same membrane is shown below the protein of interest. Circles represent number of animals: N = 8 (WT); N = 8 (*Cacna1c^+/−^*); Student’s *t*-test, * *p* < 0.05; ns = not significant. All original Western blot images from this series are shown in Appendix A.

**Figure 2 ijms-25-13713-f002:**
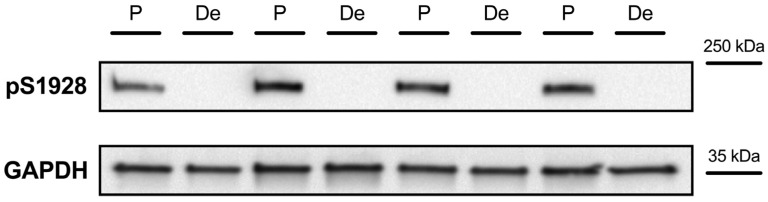
Phosphorylation of Cav1.2 at S1928 following treatment of rat hearts with a phosphorylation (P) or dephosphorylation (De) solution. Original Western blot images showing results of LV homogenates from 8 Sprague-Dawley rat hearts treated with either a phosphorylation (P) or dephosphorylation (De) solution. **Top**: results obtained with the antibody directed against phosphorylated S1928 of Cav1.2 (pS1928); **Bottom**: results obtained with the anti-GAPDH antibody. Both Western blots shown are derived from the same membrane.

**Figure 3 ijms-25-13713-f003:**
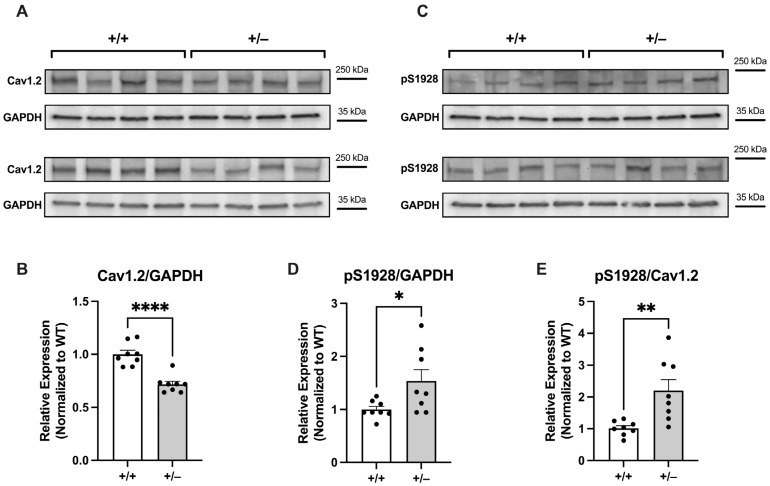
Expression of Cav1.2 and baseline phosphorylation at S1928 in LV myocardium from *Cacna1c^+/−^* and WT rats. (**A**) Original Western blot images of 8 WT (+/+) and 8 *Cacna1c^+/−^* (+/−) samples derived from two membranes probed with the anti-Cav1.2 antibody. The protein used for normalization (GAPDH) derived from the same membranes is shown below the protein of interest. (**B**) The expression of Cav1.2 (normalized to GAPDH; Cav1.2/GAPDH) is decreased by ≈30% in *Cacna1c^+/−^* (+/−). (**C**) Original Western blot images of 8 WT (+/+) and 8 *Cacna1c^+/−^* (+/−) samples derived from two membranes probed with the anti-pS1928 antibody. The protein used for normalization (GAPDH) derived from the same membranes is shown below. (**D**) Phosphorylation status of S1928 in *Cacna1c^+/−^* (+/−) versus WT (+/+) control samples when normalized to GAPDH (pS1928/GAPDH). Phosphorylation of S1928 is increased by ≈50% in *Cacna1c^+/−^* (+/−). (**E**) Phosphorylation status of S1928 in *Cacna1c^+/−^* (+/−) versus WT (+/+) control samples when normalized to Cav1.2 expression (pS1928/Cav1.2). The phosphorylation of S1928 is increased by ≈110% in *Cacna1c^+/−^* (+/−). Circles represent number of animals: N = 8 (WT); N = 8 (*Cacna1c^+/−^*); Student’s *t*-test, * *p* < 0.05, ** *p* < 0.01, **** *p* < 0.0001. Further information and all original Western blot images from this series are shown in Appendix A.

**Figure 4 ijms-25-13713-f004:**
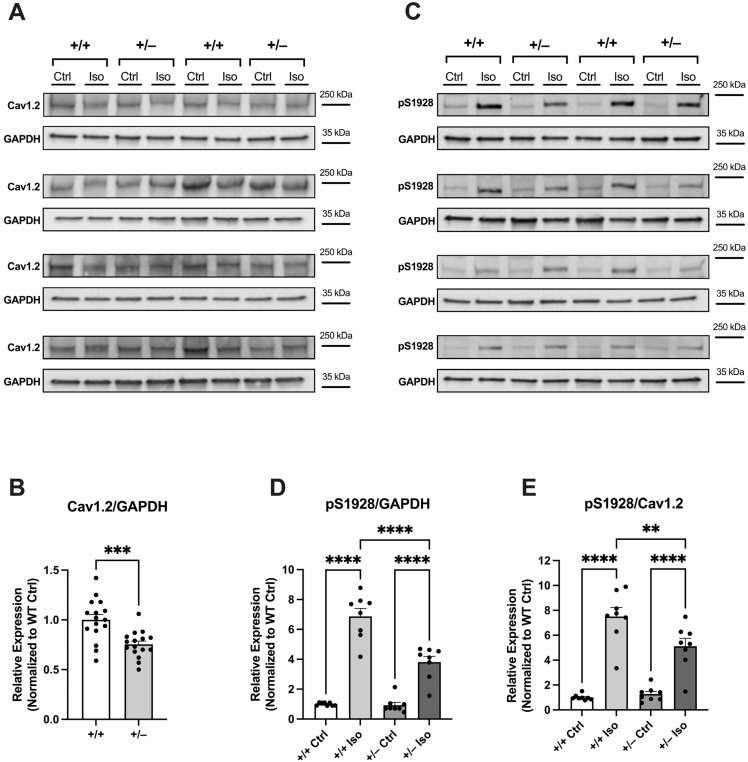
Isoprenaline-Induced Increase in Phosphorylation of Cav1.2 at S1928 in LV Myocardium from WT and *Cacna1c^+/−^* rats. (**A**) Original Western blot images of 16 WT (+/+) and 16 *Cacna1c^+/−^* (+/−) LV samples probed with anti-Cav1.2 antibody. The protein used for normalization (GAPDH) derived from the same membranes is shown below the protein of interest. Hearts were either left untreated (Ctrl) or treated with 100 nM isoprenaline (Iso). (**B**) The expression of Cav1.2 normalized to GAPDH (Cav1.2/GAPDH) was reduced in *Cacna1c^+/−^* (+/−) LV samples. (**C**) Original Western blot images of 16 WT (+/+) and 16 *Cacna1c^+/−^* (+/−) LV samples probed with anti-phospho-S1928 antibody. The protein used for normalization (GAPDH) derived from the same membranes is shown below the protein of interest. Hearts were either left untreated (Ctrl) or treated with 100 nM isoprenaline (Iso). Iso-treated samples exhibit larger phosphorylation of S1928. (**D**) Phosphorylation of S1928 normalized to GAPDH expression (pS1928/GAPDH) in the four groups: WT untreated control (+/+ Ctrl), WT treated with Iso (+/+ Iso), *Cacna1c^+/−^* untreated control (+/− Ctrl), and *Cacna1c^+/−^* treated with Iso (+/− Iso). N = 8 samples for each group. (**E**) Phosphorylation of S1928 normalized to Cav1.2 expression (pS1928/Cav1.2) in the four groups: WT untreated control (+/+ Ctrl), WT treated with Iso (+/+ Iso), *Cacna1c^+/−^* untreated control (+/− Ctrl), and *Cacna1c^+/−^* treated with Iso (+/− Iso). N = 8 samples for each group. Iso increased phosphorylation of S1928 and the Iso-induced increase was larger in WT than in *Cacna1c^+/−^* LV samples. Student’s *t*-test (**B**) or one-way ANOVA (**D**,**E**) were used for comparison of groups. ** *p* < 0.01, *** *p* < 0.001, **** *p* < 0.0001. All original Western blot images from this series are shown in Appendix A.

**Figure 5 ijms-25-13713-f005:**
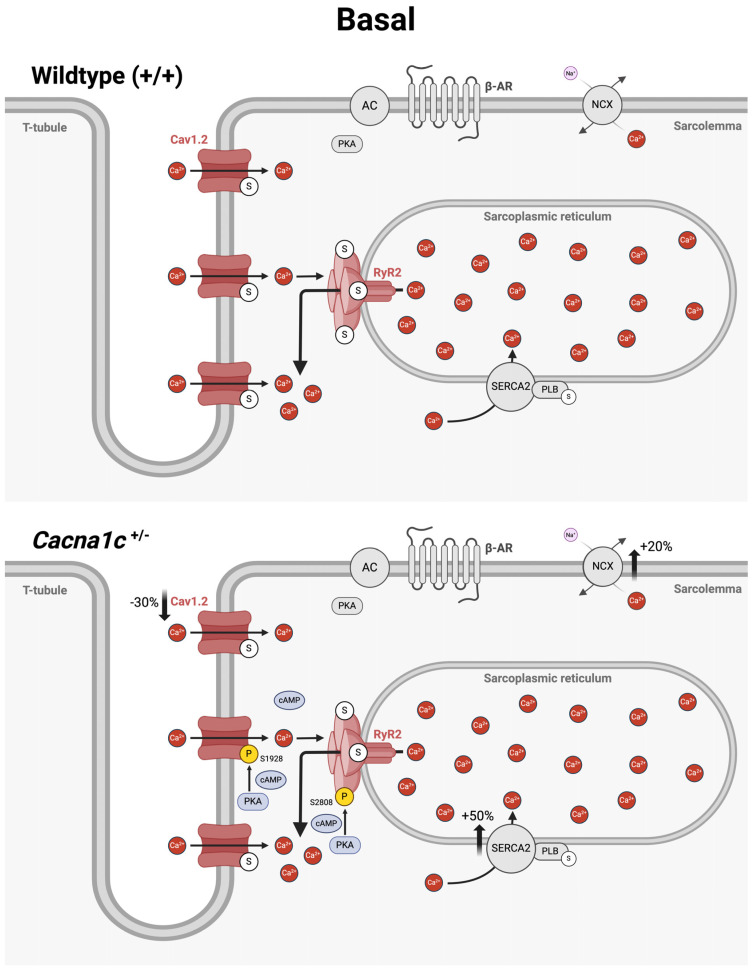
Proposed alterations of Ca^2+^ handling in ventricular myocytes from *Cacna1c^+/−^* rats under basal conditions. The scheme depicts part of a ventricular myocyte from a wildtype (**top**) and a *Cacna1c^+/−^* rat (**bottom**) with T-tubule, surface sarcolemma and the sarcoplasmic reticulum and the location of major Ca^2+^-regulating proteins: Cav1.2, RyR2, NCX, SERCA, and PLB. The space between the T-tubule (with Cav1.2) and the junctional SR (with RyR2) is termed dyadic cleft. In ventricular myocytes from *Cacna1c^+/−^* rats, the expression of Cav1.2 is reduced by 30%, whereas the expression of NCX and SERCA is elevated by +20% and +50%, respectively. β-adrenergic receptors (β-ARs) are not activated (grey), and cAMP levels in the bulk cytosol are very low. Hence, most serine (S) residues in Cav1.2, RyR2, and PLB, which are targets of PKA, are not phosphorylated, as indicated by the white circles attached to the respective proteins. In the dyadic cleft of *Cacna1c^+/−^* myocytes, however, Cav1.2 and RyR2 exhibit increased phosphorylation (P) of S1928 and S2808, respectively, as indicated by the yellow circles. The increased PKA-dependent phosphorylation of Cav1.2 and RyR2 is presumably caused by locally elevated cAMP concentration in the immediate vicinity of the channels, as depicted by the blue cAMP symbols. Abbreviations: AC, adenylate cyclase; β-AR, β-adrenergic receptor; P, phosphorylated serine residue; S, serine residue (non-phosphorylated). Created in BioRender. Königstein, D. (2024) https://BioRender.com/a97h526 (accessed on 17 December 2024).

**Figure 6 ijms-25-13713-f006:**
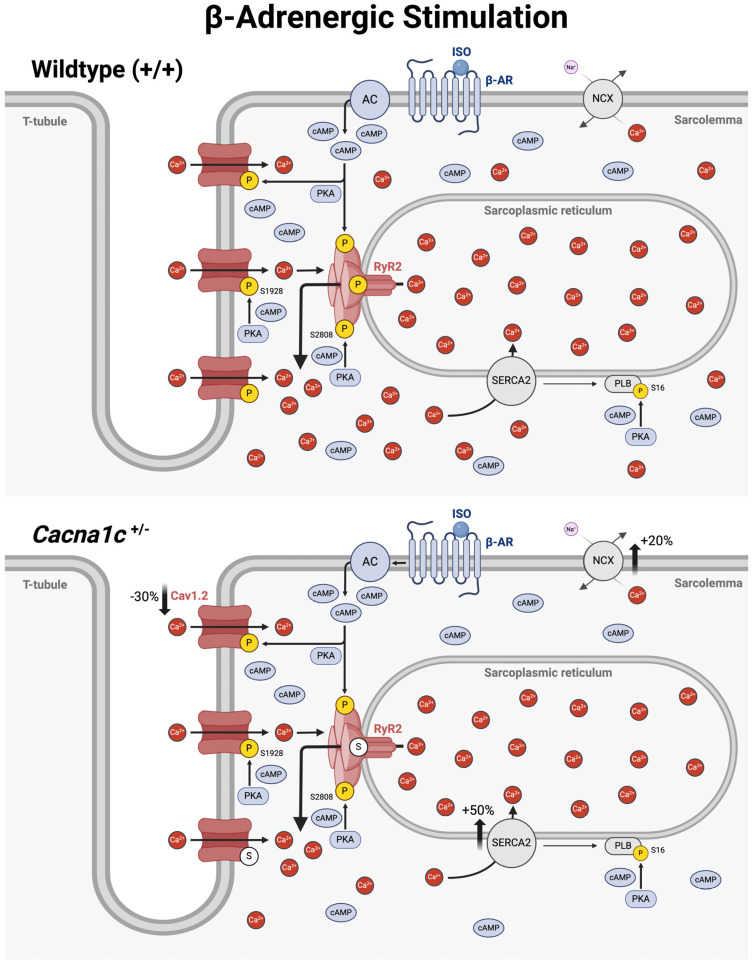
Proposed alterations of Ca^2+^ handling in ventricular myocytes from *Cacna1c^+/−^* rats during sympathetic stress. The scheme depicts part of a ventricular myocyte from a wildtype (**top**) and a *Cacna1c^+/−^* rat (**bottom**) with T-tubule, surface sarcolemma, and the sarcoplasmic reticulum and the location of major Ca^2+^-regulating proteins: Cav1.2, RyR2, NCX, SERCA, and PLB. The space between the T-tubule (with Cav1.2) and the junctional SR (with RyR2) is termed dyadic cleft. In ventricular myocytes from *Cacna1c^+/−^* rats, the expression of Cav1.2 is reduced by 30%, whereas the expression of NCX and SERCA is elevated by +20% and +50%, respectively. During the stimulation of β-AR with isoprenaline (ISO), mimicking sympathetic stress, cAMP levels increase throughout the cytosol and PKA phosphorylates target serine residues in Cav1.2 (S1928), RyR2 (S2808), and PLB (S16) in both wildtype and *Cacna1c^+/−^* myocytes. In wildtype myocytes, all available serine residues in Cav1.2 and RyR2 become phosphorylated by PKA (yellow circles). Note, however, that for *Cacna1c^+/−^* myocytes, not all available target serines (S) in Cav1.2 and RyR2 become phosphorylated by PKA, indicating a diminished phosphorylation reserve of these two channels. Abbreviations: AC, adenylate cyclase; β-AR, β-adrenergic receptor; ISO, isoprenaline; P, phosphorylated serine residue; S, serine residue (non-phosphorylated). Created in BioRender. Königstein, D. (2024) https://BioRender.com/o38g639 (accessed on 17 December 2024).

## Data Availability

All data are contained within the article or Appendix A.

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
