# Peer review of "Altered Protein Kinase A-Dependent Phosphorylation of Cav1.2 in Left Ventricular Myocardium from Cacna1c Haploinsufficient Rat Hearts"

_ijms, 2024, doi:10.3390/ijms252413713_

Round 1

Reviewer 1 Report

Comments and Suggestions for Authors

The authors present interesting data aiming to identify the mechanisms underlying the alterations of Ca2+ Handling in cardiomyocytes from  Cacna1c+/– rats. In particular, the hypothesis was that an increased phosphorylation of the calcium channel of the ammynoacidic site S1928  might explain the relative mild phenotype of these rats.

While the results might be of general interest, several questions remain open and cannot be answered by Western blot results.

Is the subcellular distribution of channel changed, thus explaining a different modulation by beta-AR subtypes (which have different distribution in TT)?

What are the effects of the basal vs. stress hyperphosphorylation, in terms of channel inactivation or current amplitude?

Did the authors tested the phosphorylation at other sites and in particular those phosphorylated by CAMK2?

Rats under 368 Basal Conditions and during Sympathetic Stress

Author Response

Reviewer #1

The authors present interesting data aiming to identify the mechanisms underlying the alterations of Ca2+ Handling in cardiomyocytes from Cacna1c+/– rats. In particular, the hypothesis was that an increased phosphorylation of the calcium channel of the ammynoacidic site S1928 might explain the relative mild phenotype of these rats.

– Reply: We would like to thank the reviewer for taking the time to evaluate our work and for providing excellent questions and comments that will help improve the paper.

Below, please find our point-by-point reply to your questions and comments.

While the results might be of general interest, several questions remain open and cannot be answered by Western blot results.

Is the subcellular distribution of channel changed, thus explaining a different modulation by beta-AR subtypes (which have different distribution in TT)?

– Reply: Thank you for raising this interesting point. Unfortunately, however, we do not know whether the subcellular distribution of the L-type Ca2+ channel (and its subcellular regulation) is changed in the Cacna1c+/- myocytes. Functional analysis of this issue would require super-resolution scanning patch clamp (e.g. Sanchez-Alonso et al. 2016. Circ Res. 119:944-955; Sanchez-Alonso et al. 2023. Circ Res. 133:120–137), a highly-sophisticated experimental technique mastered by a very limited number of laboratories and not available to us.

What are the effects of the basal vs. stress hyperphosphorylation, in terms of channel inactivation or current amplitude?

– Reply: We have measured basal and isoprenaline-stimulated sarcolemmal Ca2+ influx via L-type Ca2+ channels in our previous paper on Cacna1c+/- myocytes (Fender et al. 2023. IJMS 24, 9795). According to our results, basal L-type Ca2+ current and sarcolemmal Ca2+ influx are essentially unchanged in Cacna1c+/- myocytes versus wildtype. Sympathetic stress, via activation of beta-adrenergic receptors e.g. by isoprenaline, causes PKA-dependent phosphorylation of the channel and regulatory proteins within the channel complex with the result of an increased open probability of individual L-type Ca2+ channels and an increased whole-cell L-type Ca2+ current amplitude (McDonald et al. 1994. Physiol. Rev. 74, 365-507; van der Heyden et al. 2005. Cardiovasc. Res. 65, 28-39). In our previous paper (Fender et al. 2023. IJMS 24, 9795), we could confirm this as an increase in sarcolemmal Ca2+ influx upon isoprenaline stimulation in both Cacna1c+/- and WT myocytes. Importantly, the isoprenaline-induced increase in sarcolemmal Ca2+ influx was less pronounced in Cacna1c+/- than in WT myocytes prompting us to conduct the current study on phosphorylation of Cav1.2.

Did the authors tested the phosphorylation at other sites and in particular those phosphorylated by CAMK2?

– Reply: Cav1.2 contains multiple potential and proven phosphorylation sites for various serine/threonine kinases including PKA (van der Heyden et al. 2005. Cardiovasc. Res. 65, 28-39). Also, several potential phosphorylation sites for CaMKII have been suggested, e.g. Ser1512/1517, Ser1570/1575 or Thr1604 (Lee et al. 2006. J Biol Chem 281:25560–25567; Wang et al. 2009. J Physiol Sci (2009) 59:283–290). We would love to study the CaMKII-dependent phosphorylation of Cav1.2 in our model, in particular because we have found altered isoprenaline-mediated phosphorylation of the ryanodine receptor, RyR2, at the CaMKII site Ser2814 in Cacna1c+/- myocytes in our previous study (Fender et al. 2023. IJMS 24, 9795). Unfortunately, however, to the best of our knowledge, there is no antibody available that would detect reliably a proven CaMKII-specific phosphorylation site in the cardiac Cav1.2.

Reviewer 2 Report

Comments and Suggestions for Authors

Major Comments

1.       Western blot against S1928 in Cav1.2 and Cav1.2 in Figure 2-4, but the 250 kDa heights shown in the figures are all different. s1928 is a serine within Cav1.2 and the height at detection should be the same as in Cav1.2. The molecular weight markers should be accurately indicated.

2.       The expression levels of Cav1.3 and Cavβ2 in Figure 1 and Cav1.2 in Figure 3.4, Cav1.2 is expressed in all cardiomyocytes while Cav1.3 is expressed specifically in the atria (Matteo E. Mangoni, et al. PNAS April 16, 2003, 100 (9) 5543-55482014 Jan 21;3(2):15-38). The authors have examined the expression of Cav1.3 in extracts from left ventricular tissue, are these results reproducible? Other factors that should also be examined are action potential shifts and mRNA expression levels. The mechanisms underlying changes in expression levels should be clarified. The mechanisms underlying the changes in expression levels should be clarified.

3.       In their discussion, the authors focus on cAMP levels and PKA activity as the cause of decreased Cav1.2 expression. cAMP produced by β-adrenergic receptor stimulation activates Epac (exchange proteins directly activated by cAMP) in addition to PKA. This signal activates calcium reuptake in the endoplasmic reticulum (ER). This signal is involved in the regulation of calcium reuptake in the endoplasmic reticulum (Okumura S, et al. J Clin Invest. 2014;124:2785-2801.). In other words, it is questionable whether the experimental system conducted by the authors is able to represent PKA-dependent phosphorylation; consideration of CaMKII and Ca2+-induced Ca2+ release (CICR) mechanisms may also be necessary.

Minor Comments

There are numerous errors in the manuscript. For example, the abstract has too many words. The abstract is a single paragraph of about 200 words maximum The authors should carefully check the text again.

Author Response

Reviewer #2

The authors would like to thank the reviewer for the constructive questions and comments that will help improve the paper.

Below, please find our point-by-point reply to your questions and comments.

Major Comments

  1. Western blot against S1928 in Cav1.2 and Cav1.2 in Figure 2-4, but the 250 kDa heights shown in the figures are all different. s1928 is a serine within Cav1.2 and the height at detection should be the same as in Cav1.2. The molecular weight markers should be accurately indicated.

– Reply: The molecular weight markers shown in the figures are accurate. The protein ladder that we use contains a molecular weight marker at 250 kDa, which is shown in each of the figures. Indeed, the total Cav1.2 protein detected by the Alomone antibody (#ACC-003) and the Cav1.2 protein phosphorylated at S1928 and detected by the Invitrogen/ThermoFisher antibody (#PA5-64748) migrate at different molecular weights. The anti-Cav1.2 antibody detects a band at close to 250 kDa, whereas the anti-pS1928 antibody detects a band that migrates at somewhat below 200 kDa. The reason for this discrepancy is unknown. However, a similar observation has been made before by others in cardiac tissue. In particular, a most recent paper on Cav1.2 in human atrial myocardium (Grammatika Pavlidou et al. 2023. Eur. Heart J. (2023) 44, 2483–2494) found that the PKA-phosphorylated Cav1.2 at S1928 (using a different anti-pS1928 antibody) migrated at a similar molecular weight as we have found in our study.

  1. The expression levels of Cav1.3 and Cavβ2 in Figure 1 and Cav1.2 in Figure 3.4, Cav1.2 is expressed in all cardiomyocytes while Cav1.3 is expressed specifically in the atria (Matteo E. Mangoni, et al. PNAS April 16, 2003, 100 (9) 5543-55482014 Jan 21;3(2):15-38). The authors have examined the expression of Cav1.3 in extracts from left ventricular tissue, are these results reproducible? Other factors that should also be examined are action potential shifts and mRNA expression levels. The mechanisms underlying changes in expression levels should be clarified. The mechanisms underlying the changes in expression levels should be clarified.

– Reply: In adult heart, Cav1.3 is expressed predominantly in supra-ventricular (atrial) tissue and not or only marginally so in the ventricles (Zaveri et al. 2023. Front. Physiol. 14:1144069). It is clear, however, that Cav1.3 expression is regulated in a developmental manner: Cav1.3 exhibits high expression in fetal and neonatal heart, and then, in the adult heart, its expression declines particularly in the ventricles, whereas its expression remains rather large in the atria (Qu et al. 2011. Pediatr. Res. 69, 479-85). With pathology, however, Cav1.3 expression may return in ventricular tissue. In human heart failure, Cav1.3 expression has been demonstrated in left ventricular tissue from patients with ischemic cardiomyopathy (Srivastava et al. 2020. Heart Rhythm 17, 1193-97). Moreover, it appears that Cav1.3 expression in the ventricles returns in older adult heart. In ventricles from 12- and 24-months-old mice, clear evidence for expression of Cav1.3 has been obtained (Walton et al. 2016. J Gerontol A Biol Sci Med Sci. 71, 1005–1013). Since the rats in our study were also rather old (9-15 months), we assume that this may have led to a detectable expression of Cav1.3 in ventricular tissue.

We did not aim at measuring mRNA levels of Cav1.3, as it is well-known that altered mRNA levels do not necessarily translate into the same changes on the protein level. Moreover, we did not try to elucidate the mechanisms underlying the expression changes of Cav1.3 in our Cacna1c+/- model, since this was not relevant for the question of altered regulation of PKA-dependent Cav1.2 phosphorylation that we pursued. Whatever the cause of the detection and alterations of Cav1.3 expression in the ventricles of the Cacna1c+/- rats, the results clearly demonstrate that the Cacna1c+/- ventricles do not exhibit a higher Cav1.3 expression than the WT controls, thus excluding the possibility that a compensatory increase in Cav1.3 expression could explain the unaltered L-type Ca2+ current and sarcolemmal Ca2+ influx observed in the Cacna1c+/- ventricular myocytes.

  1. In their discussion, the authors focus on cAMP levels and PKA activity as the cause of decreased Cav1.2 expression. cAMP produced by β-adrenergic receptor stimulation activates Epac (exchange proteins directly activated by cAMP) in addition to PKA. This signal activates calcium reuptake in the endoplasmic reticulum (ER). This signal is involved in the regulation of calcium reuptake in the endoplasmic reticulum (Okumura S, et al. J Clin Invest. 2014;124:2785-2801.). In other words, it is questionable whether the experimental system conducted by the authors is able to represent PKA-dependent phosphorylation; consideration of CaMKII and Ca2+-induced Ca2+ release (CICR) mechanisms may also be necessary.

– Reply: In the heart, beta-adrenergic receptors – via Gs proteins – stimulate adenylate cyclase activity to increase cAMP levels. cAMP, in turn, stimulates PKA activity and PKA phosphorylates various target proteins to alter cardiomyocyte function. One of these target proteins is phospholamban (PLB) with Ser16 as the major target site for PKA phosphorylation. This pathway as well as the PKA target site in PLB have been established for decades and have been the topic of numerous original and review articles (e.g. MacLennan & Kranias, 2003. Nat. Rev. Mol. Cell Biol. 4, 566-77; Kranias & Hajjar, 2012. Circ. Res. 110, 1646-1660; Papa et al. 2022. Annu. Rev. Physiol. 2022. 84:285–306). More recently, it became clear that cAMP may not only signal via PKA but also via Epac. The two pathways are not mutually exclusive but rather represent divergent pathways for cAMP signaling expected to be activated in parallel within a given cell. The nice work mentioned above by the reviewer (Okumura et al. 2014. J. Clin. Invest. 124:2785-2801) deals with the role of Epac in calcium signaling in mouse hearts/cardiomyocytes. The authors show – mainly by using Epac1 KO mice – that Epac1 can regulate Ser16 phosphorylation of PLB by a phospholipase C-PKC-dependent pathway. They do not rule out, however, that Ser16 can also be phosphorylated by PKA. In fact, their data show directly that – even in Epac1 KO mice – isoprenaline still greatly increases Ser16 phosphorylation of PLB (Fig. 3A in Okumura et al. 2014). Moreover, the authors state explicitly that the proposed Epac pathway acts in addition to the PKA pathway: "Our current study indicates that PLN on serine-16 and RyR2 on serine-2808 and serine-2814 are phosphorylated by EPAC1 in addition to and independently of PKA or CaMKII." (from the Legend of Fig. 9 in Okumura et al. 2014). Thus, our experimental system is clearly able to represent PKA-dependent phosphorylation as shown by the isoprenaline-induced increases in known PKA sites in various target proteins including Ser16 in PLB, S2808 in RyR2 and S1928 in Cav1.2. Of particular note, the experimental system used by Okumura et al. demonstrating isoprenaline-induced increases in Ser16 phosphorylation of PLB in the absence of Epac1 (Epac1 KO mice) is very much comparable to our system (they used Langendorff-perfused mouse hearts, 100 nM isoprenaline for 5 minutes – we used Langendorff-perfused rat hearts, 100 nM isoprenaline for 5 minutes). Finally, we also considered CaMKII and CICR mechanisms to explain our results. In fact, our previous study (Fender et al. 2023. IJMS 24, 9795) has demonstrated that isoprenaline-induced phosphorylation of Ser2814 (CaMKII site) in RyR2 is altered in Cacna1c+/- hearts and that CICR mechanisms involving Ca2+ influx via L-type Ca2+ channels and Ca2+ release via RyR2 must likewise be altered since the phosphorylation of both Cav1.2 (this study) and RyR2 (previous study) are altered and the isoprenaline-induced increase in sarcolemmal Ca2+ influx and Ca2+ transients is impaired in Cacna1c+/- myocytes (Fender et al. 2023. IJMS 24, 9795). The latter issue, i.e. impaired CICR during beta-adrenergic stimulation in Cacna1c+/- myocytes, is explained in detail in the Discussion, Chapter 3.2.

Minor Comments

There are numerous errors in the manuscript. For example, the abstract has too many words. The abstract is a single paragraph of about 200 words maximum The authors should carefully check the text again.

– Reply: We have changed the abstract. It now contains 200 words. We have also checked carefully the remainder of the manuscript. We are not aware of any more errors.

Reviewer 3 Report

Comments and Suggestions for Authors

Reviewing a manuscript entitled, “Altered Protein Kinase A-Dependent Phosphorylation of Cav1.2 in Left Ventricular Myocardium from Cacna1c Haploinsufficient Rat Hearts” by Königstein D, et al., this is an article focusing on role of Cav1.2 in sympathetic stress responses in cardiomyocytes from haploinsufficient Cacna1c (Cacna1c+/–) rats. In a painstaking experiment using the Langendorff experimental system, the authors well explained the changes in the function of Cacna1c+/– cardiomyocytes by the expression level and hyperphosphorylation of Cav1.2. However, there is no mention of the mechanisms of them. Therefore, the authors should address the following concerns.

 The authors should provide data on the cardiac function of Haploinsufficient Cacna1c (Cacna1c+/–) rats under basal and catecholamine stimulation.

For example, are Haploinsufficient Cacna1c (Cacna1c+/–) rats less able to handle exercise stress than controls?

 It is impressive that the functional changes of Cacna1c+/– cardiomyocytes are explained by the expression level and hyperphosphorylation of Cav1.2 through painstaking experiments using the Langendorff experimental system. The authors should mention in the discussion the molecular biological mechanisms of these characteristics of Cacna1c+/– cardiomyocytes.

 The authors should explain in more detail how the phenomenon occurring in cardiomyocytes of Cacna1c+/– rats is a model case of human heart failure. Also, does the human heart failure the authors are talking about refer to HFrEF or does it also apply to HFpEF?

 Figure 5 is a wonderful and easy-to-understand diagram. If possible, it would be even easier to understand if the authors make it basal state A-1 (+/+), A-2 (+/-), catecholamine-stimulated B-1 (+/+), B-2 (+/-).

Author Response

Reviewer #3

The authors would like to thank the reviewer for the encouraging and constructive questions and comments that will help improve the paper.

Below, please find our point-by-point reply to your questions and comments.

Reviewing a manuscript entitled, “Altered Protein Kinase A-Dependent Phosphorylation of Cav1.2 in Left Ventricular Myocardium from Cacna1c Haploinsufficient Rat Hearts” by Königstein D, et al., this is an article focusing on role of Cav1.2 in sympathetic stress responses in cardiomyocytes from haploinsufficient Cacna1c (Cacna1c+/–) rats. In a painstaking experiment using the Langendorff experimental system, the authors well explained the changes in the function of Cacna1c+/– cardiomyocytes by the expression level and hyperphosphorylation of Cav1.2. However, there is no mention of the mechanisms of them. Therefore, the authors should address the following concerns.

The authors should provide data on the cardiac function of Haploinsufficient Cacna1c (Cacna1c+/–) rats under basal and catecholamine stimulation.

For example, are Haploinsufficient Cacna1c (Cacna1c+/–) rats less able to handle exercise stress than controls?

– Reply: Indeed, Cacna1c+/- rats, at least when examining their hearts or isolated ventricular myocytes, are less able to handle stress with regard to functional changes (Ca2+ transients, sarcomere shortenings) or phosphorylation of Ca2+ handling proteins (Cav1.2, RyR2) induced by isoprenaline, as shown in this study and in our previous study (Fender et al. 2023. IJMS 24, 9795). These documented cellular changes would imply that in vivo the Cacna1c+/- rats could show an impaired cardiac response to exercise stress. We do not have any such in vivo data, however, so that we can not answer this question directly with experimental data.

It is impressive that the functional changes of Cacna1c+/– cardiomyocytes are explained by the expression level and hyperphosphorylation of Cav1.2 through painstaking experiments using the Langendorff experimental system. The authors should mention in the discussion the molecular biological mechanisms of these characteristics of Cacna1c+/– cardiomyocytes.

– Reply: In the Discussion, we mention the molecular mechanisms underlying the regulation of Cav1.2 and its consequences for cellular Ca2+ signaling of Cacna1c+/- myocytes in several chapters: In chapter 3.1 we explain the mechanisms under basal conditions, and in chapter 3.2 the mechanisms during sympathetic stimulation. Moreover, in chapter 3.3 we now explain in more detail how the mechanisms in Cacan1c+/- myocytes resemble the situation in human heart failure with reduced ejection fraction. Finally, in new Figures 5 and 6 we now illustrate the molecular changes in Cacna1c+/- myocytes with regard to PKA-dependent phosphorylation of Cav1.2 and RyR2 in more detail, and this is explained in the accompanying text.

The authors should explain in more detail how the phenomenon occurring in cardiomyocytes of Cacna1c+/– rats is a model case of human heart failure. Also, does the human heart failure the authors are talking about refer to HFrEF or does it also apply to HFpEF?

– Reply: The human heart failure we are talking about refers to HFrEF (heart failure with reduced ejection fraction). We now mention and clarify this in the Discussion, Chapter 3.3.

Not all of the aspects of Ca2+ handling remodeling occurring in Cacna1c+/- myocytes are similar to the situation in human HFrEF, and we do not want to imply that Cacna1c+/- myocytes in general are a model case of human HFrEF. For example, SERCA2a is up-regulated in Cacna1c+/- myocardium, whereas it is usually down-regulated in human HFrEF. Remarkably, however, many similarities exist with regard to PKA-dependent phosphorylation/regulation of Cav1.2 and RyR2, which both appear to be hyperphosphorylated in Cacna1c+/- myocardium and in human HFrEF. We now elaborate in more detail on this in the Discussion, Chapter 3.3.

Figure 5 is a wonderful and easy-to-understand diagram. If possible, it would be even easier to understand if the authors make it basal state A-1 (+/+), A-2 (+/-), catecholamine-stimulated B-1 (+/+), B-2 (+/-).

– Reply: Thank you very much for this suggestion. We have altered the scheme according to your suggestion and now differentiate between basal state (Wildtype vs Cacna1c+/-) and beta-adrenergic stimulation (Wildtype vs Cacna1c+/-). Because the figure has grown much larger through the additions, we have split it in two and now present the basal state as new Figure 5 and the situation during beta-adrenergic stimulation as new Figure 6. The figure legends and the text in the Discussion have been changed accordingly.

Round 2

Reviewer 1 Report

Comments and Suggestions for Authors

No further comments, thanks

Reviewer 2 Report

Comments and Suggestions for Authors

well responded to my comments